# SARS-CoV-2 Neutralization Assays Used in Clinical Trials: A Narrative Review

**DOI:** 10.3390/vaccines12050554

**Published:** 2024-05-18

**Authors:** Yeqing Sun, Weijin Huang, Hongyu Xiang, Jianhui Nie

**Affiliations:** 1School of Life Sciences, Jilin University, Changchun 130012, China; yeqing22@mails.jlu.edu.cn; 2Division of HIV/AIDS and Sex-Transmitted Virus Vaccines, National Institutes for Food and Drug Control, State Key Laboratory of Drug Regulatory Science, NHC Key Laboratory of Research on Quality and Standardization of Biotech Products, NMPA Key Laboratory for Quality Research and Evaluation of Biological Products, Beijing 102629, China; huangweijin@nifdc.org.cn

**Keywords:** SARS-CoV-2, neutralizing antibodies, clinical trials, standards, correlation of protection

## Abstract

Since the emergence of COVID-19, extensive research efforts have been undertaken to accelerate the development of multiple types of vaccines to combat the pandemic. These include inactivated, recombinant subunit, viral vector, and nucleic acid vaccines. In the development of these diverse vaccines, appropriate methods to assess vaccine immunogenicity are essential in both preclinical and clinical studies. Among the biomarkers used in vaccine evaluation, the neutralizing antibody level serves as a pivotal indicator for assessing vaccine efficacy. Neutralizing antibody detection methods can mainly be classified into three types: the conventional virus neutralization test, pseudovirus neutralization test, and surrogate virus neutralization test. Importantly, standardization of these assays is critical for their application to yield results that are comparable across different laboratories. The development and use of international or regional standards would facilitate assay standardization and facilitate comparisons of the immune responses induced by different vaccines. In this comprehensive review, we discuss the principles, advantages, limitations, and application of different SARS-CoV-2 neutralization assays in vaccine clinical trials. This will provide guidance for the development and evaluation of COVID-19 vaccines.

## 1. Introduction

The severe acute respiratory syndrome coronavirus 2 (SARS-CoV-2) was first reported in Wuhan, China, in December 2019 [1], followed by the pandemic of coronavirus disease 2019 (COVID-19) [2,3,4]. After vaccination or viral infection, the innate/non-specific immune system is initially activated. Antigen-presenting cells, including macrophages and dendritic cells, engulf the exogenous antigens. Subsequently, antigen-presenting cells transmit antigen information to T cells and B cells, activating the adaptive/specific immune system. Activated CD4+ T cells secrete various cytokines to further stimulate immune responses, while CD8+ T cells eliminate virus-infected cells [5]. B cells are also activated to produce specific antibodies, including neutralizing antibodies (nAbs) and binding antibodies [6]. Therefore, nAbs, cellular immunity, innate immunity, and other immunological indicators are vital in assessing the effectiveness of viral vaccines [7,8]. The importance of measuring nAbs in COVID-19 lies in the fact that they can serve as correlates of protection. Quantitative detection of nAbs specific to SARS-CoV-2 facilitates monitoring of viral infections. Furthermore, nAbs can be used to assess monoclonal antibodies, convalescent sera, and vaccine effectiveness [9]. Therefore, the detection of nAbs against SARS-CoV-2 is important in clinical trial investigations.

On 5 May 2023, the World Health Organization (WHO) announced that COVID-19 no longer constituted a public health emergency of international concern [10]. However, this announcement does not signify the end of COVID-19. Research and development efforts regarding vaccines against SARS-CoV-2 are ongoing, especially for broadly protective candidate vaccines. Therefore, continuous monitoring is necessary to track emerging variants of SARS-CoV-2 and to facilitate the development of the next-generation vaccines. In this review, we searched PubMed from January 2020 to 29 April 2024, to identify all the eligible studies in order to comprehensively describe the progress in SARS-CoV-2 neutralization assays used in clinical trials.

## 2. Current Status of SARS-CoV-2 Vaccines

Since the first COVID-19 vaccine on 22 July 2020, as of February 2024, a total of about 13.59 billion vaccine doses have been administered worldwide [11]. The types of COVID-19 vaccines mainly include inactivated vaccines, recombinant protein subunit vaccines, viral vector vaccines, and nucleic acid vaccines (Table 1).

Cytopathic effect (CPE). Plaque reduction neutralization test (PRNT). Microneutralization (MN). Geometric mean titer (GMT). Geometric mean concentration (GMC).

The safety and efficacy of the aforementioned vaccines have been extensively validated globally [12,13,14,15,16,17,18,19,20,21,22,23,24,25,26,27,28,29,30,31,32,33,34,35]. However, the effectiveness of these vaccines may vary among different populations and in the presence of variant viruses. Because different types of vaccines have been subject to different methods during their research and development phases, it is impossible to compare them directly. Nevertheless, all vaccines that have been listed for emergency use by the WHO are proven to be highly effective in preventing severe COVID-19 illness.

As a conventional platform, inactivated vaccines have long played a critical role in the battle against infectious diseases [36]. Physical or chemical methods are used to inactivate the pathogen, rendering it non-infectious and non-pathogenic while retaining its immunogenicity. Some inactivated vaccines are formulated with appropriate adjuvants to further enhance their efficacy [37]. Two examples of inactivated COVID-19 vaccines are BBIBP-CorV from the Beijing Institute of Biological Products and CoronaVac from Sinovac Biotech. The inactivated vaccine developed by the Wuhan Institute of Biological Products, a subsidiary of the China National Pharmaceutical Group (Sinopharm), uses the SARS-CoV-2 WTV04 strain [16]. Sinovac’s inactivated vaccine, CoronaVac, uses the SARS-CoV-2 CN02 strain and was officially listed on the World Health Organization’s Emergency Use Listing in June 2021 [12]. Inactivated vaccines have shown good safety profiles, high immunogenicity, and minimal adverse reactions [38].

Recombinant protein subunit vaccines rely on the expression of protective antigens in prokaryotic or eukaryotic cells and are usually formulated with appropriate adjuvants. ZF2001, a recombinant protein vaccine from Chongqing Zhifei Biological Products, uses the Chinese hamster ovary (CHO) cell line to express purified SARS-CoV-2 S-RBD dimeric antigen protein and is formulated with alum as an adjuvant [34]. NVX-CoV2373, a recombinant protein vaccine from Novavax, is based on a nanoparticle containing the full-length S protein plus Matrix-M adjuvant. Subunit vaccines have high purity, stability, and safety profiles and are generally administered in a two-dose regimen [39].

Viral vector vaccines are constructed by inserting specific antigenic nucleotide sequences into viral vectors to express the antigens of interest in host cells. Ad5-nCoV, a human type 5 adenovirus vector vaccine from China, is the first viral vector vaccine to enter clinical trials [40]. This vaccine induces sustained humoral and cellular immune responses with a single-dose injection [41]. Another viral vector vaccine is AZD1222 (ChAdOx1 nCoV-19) developed by the University of Oxford/AstraZeneca, which uses a replication-deficient chimpanzee adenovirus vector to carry the SARS-CoV-2 spike (S) gene. After vaccination, the S protein expressed in vivo can trigger the immune system to generate antibodies and cellular immune responses against SARS-CoV-2 [42]. Sputnik V is one of the first COVID-19 vaccines approved for emergency use, developed by the Gamaleya Research Center in Russia, and it is the first non-Western vaccine to complete all phase III clinical trials [43]. Sputnik V uses two different adenoviral vectors (rAd26 and rAd5) to transmit the S protein gene of the virus, which can enhance the immune system’s response and improve the effectiveness and duration of the vaccine [44]. Multiple studies have demonstrated that Sputnik V is highly effective in preventing both symptomatic and severe infections of COVID-19.

Nucleic acid vaccines (DNA or RNA) are designed to deliver genes that encode protective antigens into host cells and use the host translation machinery to produce the antigen protein and induce immune responses against the target pathogen [45]. mRNA vaccines have the advantage of a shorter development cycle, flexible design, and the ability to expand the range of involved antigens for an enhanced breadth of immune responses. Moreover, the exogenous mRNA itself can act as a self-adjuvant, eliminating the need for additional adjuvants [46,47]. Compared with mRNA, DNA molecules are stable and can persist in host cells for a long time, continuously generating endogenous antigens to elicit a sustained immune response [48]. However, owing to the poor delivery efficiency of DNA vaccines, relatively lower immune responses are induced in comparison with mRNA vaccines.

To address the issue of immune evasion by SARS-CoV-2 variants, a variety of approaches have been used to enhance vaccine efficacy against the circulating strains. On 31 August 2022, the U.S. Food and Drug Administration (FDA) authorized the Pfizer-BioNTech and Moderna bivalent COVID-19 vaccines for use as booster shots [49]. Moderna’s bivalent vaccines, mRNA-1273.222 (targeting the original S protein and Omicron BA.4/5 variant) and mRNA-1273.214 (targeting the original S protein and Omicron variant), use a mixture of mRNAs encoding the Omicron S protein and the Wuhan-1 S protein [23,50]. Clinical trials have demonstrated that, as booster shots, the mRNA-1273.214 and mRNA-1273.222 bivalent vaccines have safety profiles similar to that of the monovalent mRNA-1273 SARS-CoV-2 vaccine and exhibit superior neutralizing antibody responses against the Omicron variant compared with mRNA-1273 used as a booster [19]. Following a similar strategy, Pfizer-BioNTech developed its bivalent SARS-CoV-2 vaccine, which encodes the S protein of both the original SARS-CoV-2 strain and the Omicron BA.4/BA.5 variant. When used as a booster, the protective efficacy against symptomatic SARS-CoV-2 infection of Pfizer-BioNTech’s bivalent SARS-CoV-2 vaccine is enhanced by 28% to 56% compared with the original vaccine [51]. Various studies have demonstrated that bivalent COVID-19 vaccines can maintain immune protection against the original virus strain while triggering a specific neutralizing antibody response against the currently prevalent variant, indicating a broader spectrum of protection. However, with the continuing evolution of SARS-CoV-2 and changing antigenicity of emerging variants, the WHO, European Medicines Agency, and FDA have all recommended updating monovalent XBB component-containing COVID-19 vaccines for the autumn-winter season of 2023/2024.

## 3. Efficacy of SARS-CoV-2 Vaccines

All the approved vaccines on the WHO emergency use list must show a minimum efficacy of 50% or higher in clinical trials [52]. After approval, their safety and efficacy are continuously monitored. The clinical evaluation indicators for SARS-CoV-2 vaccines include the aspects of safety, immunogenicity, and effectiveness. Phase I/II trials typically focus on assessing the safety and immunogenicity of the vaccine, whereas Phase III trials primarily aim to verify its safety and effectiveness in large populations. The evaluation of vaccine effectiveness involves multiple dimensions of human immune system involvement.

Upon vaccination, the human body can generate neutralizing antibodies (nAbs) against SARS-CoV-2, which confer protective immunity [53]. These nAbs constitute only a small portion of the antibodies secreted by B cells yet possess antiviral activity. The titer of nAbs in the blood is crucial, representing a key indicator in evaluating the effectiveness of SARS-CoV-2 vaccines.

The gold standard for assessing vaccine effectiveness is the relative protection rate, which requires large-scale clinical trials. However, clinical trials are time-consuming and often struggle to keep pace with the speed of viral mutations. Currently, most studies suggest that vaccines primarily induce nAbs, leading to the possibility of using nAbs as surrogate endpoints. A surrogate endpoint refers to a biomarker that can be used as a substitute for the clinical endpoint. The detection of nAbs can effectively save time in vaccine clinical trials as compared with large-scale clinical trials [54].

## 4. Role of Neutralizing Antibodies

Vaccination or natural infection induces the production of cellular and humoral immunity in the body, generating antibodies through immune synergistic effects to protect against viral infections [55]. Studies have shown that different levels of antibody, including nAbs and binding antibodies, can be produced 1–2 weeks after vaccination or natural infection, with only a small portion of these antibodies being nAbs [53]. As a special type of antibody produced by B lymphocytes, nAbs can bind to the surface of viral particles, thereby blocking the viral replication cycle and preventing viral infection of cells [56]. As crucial immune markers, the mechanisms of action of nAbs generally include: (1) altering the viral surface configuration; (2) binding to virus-binding sites involved in adsorption to prevent viral attachment and cell invasion; (3) forming immune complexes with the virus, which can be easily engulfed and cleared by macrophages; and (4) activating the complement system upon binding to the surface antigens of enveloped viruses, leading to viral dissolution [57,58]. Currently, several approaches to controlling SARS-CoV-2 are being evaluated in clinical trials, including vaccines, monoclonal antibody therapy, and convalescent plasma, all of which rely on nAbs to achieve preventive or therapeutic effects [59].

SARS-CoV-2 has four structural proteins: spike (S), envelope (E), membrane (M), and nucleocapsid (N) proteins. The virus enters host cells by binding the receptor-binding domain (RBD) of the S protein to the angiotensin-converting enzyme 2 (ACE2) receptor on the surface of host cells [60]. The nAbs against SARS-CoV-2 can bind to the viral surface S protein and prevent its binding to the ACE2 receptor on host cells, thereby inhibiting viral infection [61]. Studies have shown a correlation between the level of nAbs against SARS-CoV-2 and the level of vaccine protection [62,63]. Neutralizing antibody assays can be used to determine indicators, such as the seroconversion rate of nAbs and the average antibody titer, which help assess the presence and level of nAbs in serum and evaluate the immunological efficacy of vaccines [64].

## 5. Detection Methods for Neutralizing Antibodies

Neutralizing antibody detection methods can be classified into three main categories: the live (conventional) virus neutralization test (cVNT), pseudovirus neutralization test (pVNT), and surrogate virus neutralization test (sVNT). Among them, the cVNT is the traditional method and is considered the gold standard for evaluating neutralizing antibody capability. The mechanism of this method involves mixing the antibodies to be tested with virus-infected cells and observing whether they can neutralize the virus and prevent infection. This method is based on live viruses, requires a high biosafety level (BSL-3), involves operational risks, and has a relatively long testing period. These factors pose challenges and limit the application of this approach in vaccine, drug, and antibody research [65] The pVNT method is an alternative to the cVNT that follows a similar mechanism. It uses a pseudovirus that can only infect once without replication. This method overcomes the limitations of traditional cVNT methods in terms of the experimental facility and equipment needed. The pVNT can be conducted in a BSL-2 laboratory, has a relatively shorter testing time, and provides greater possibilities for antibody detection, vaccine research, and drug screening. This approach can be used to combat emerging and re-emerging viruses in a more effective manner [66].

The sVNT is a non-virus neutralization antibody detection method based on molecular interactions. It does not require any viruses or cells for testing. The sVNT can be conducted in laboratories with lower biosafety level standards; it is simple and rapid, and some of those measures showed good consistency with results obtained from live virus neutralization assays (cVNT) [67].

## 6. Live (Conventional) Virus Neutralization Test (cVNT)

The live virus-neutralizing antibody detection methods are considered the gold standard for neutralizing antibody detection, which is a vital part of the immunogenicity analysis of the clinical trials for candidate vaccines. The main methods include the plaque reduction neutralization test (PRNT) [68] and the cytopathic effect (CPE) assay [69]. Both methods involve mixing quantified live viruses with different dilutions of serum, inoculating the mixture onto a prepared cell monolayer, and evaluating the degree of cell damage as an indicator of neutralizing antibody titer (Figure 1). Additionally, recombinant live virus detection methods with reporter genes have been employed and shown advantages in SARS-CoV-2 neutralizing antibody detection [70].

The cVNT remains the standard and classical method for evaluating nAbs in virus vaccine evaluation. It is methodologically reliable and serves as a reference for evaluating all other neutralizing antibody detection methods. However, all cVNT methods rely on the use of infectious live viruses, which restricts the testing to BSL-3/4 laboratory environments for some viruses causing severe diseases. This greatly limits the number of laboratories capable of conducting these assays. Additionally, owing to its low throughput, live virus-based neutralizing antibody detection methods are not suitable for large-scale sero-epidemics study and vaccine evaluation [71].

### 6.1. Cytopathic Effect (CPE)

The cytopathic effect (CPE) assay is a traditional method used to quantify viruses. CPE refers to the changes in cell morphology, structure, and function that occur when cells are infected by a virus, which ultimately leads to cell death or lysis [72]. The CPE method can be used in neutralizing antibody detection to assess the neutralizing effect of antibodies against the virus. In the laboratory, the CPE method typically involves mixing the virus with the corresponding nAbs and adding them to a host cell culture. As the virus replicates and spreads, changes in cell morphology and function can be observed. If the nAbs are effective, viral replication and spread will be inhibited, resulting in a reduction or elimination of CPE [73]. For example, in phase II clinical trials of COVID-19 inactivated vaccines such as CoronaVac and BBIBP-CorV, the CPE method was used for neutralizing antibody detection [12,16].

### 6.2. Plaque Reduction Neutralization Test (PRNT)

The plaque reduction neutralization test (PRNT) is considered the gold standard for measuring SARS-CoV-2 nAbs [74]. It is a highly specific method that can detect the cross-reactivity of nAbs against different viral strains or variants. PRNT is an important tool for the development and evaluation of novel antiviral drugs or vaccines in preclinical and clinical studies [75].

The PRNT is a neutralization assay based on the quantification of viral plaques formed as a result of virus-induced cytopathic effects (CPE) in cells. Briefly, cells are seeded in a multi-well plate, and the test plasma or serum samples are mixed with different dilutions of the virus solution. The mixture is then used to infect host cells. After 1 h of incubation, the culture medium is removed, and a semi-solid medium containing agarose or carboxymethylcellulose is overlayed to prevent further uncontrolled virus spread. Infected host cells undergo lysis and infect neighboring cells, resulting in a cycle of infection and lysis, ultimately forming viral plaques. These plaques can be directly observed using the naked eye or an optical microscope. Typically, 4 days are needed for SARS-CoV-2 to form visible plaques (plaque-forming units, PFU) [76]. The number of plaques is manually counted, with the assumption that each plaque arises from a single infectious virus particle. Therefore, PFU/mL represents the number of infectious virus particles per milliliter of the tested sample and corresponds to the viral infectivity level determined using titration evaluation [77,78]. If the plasma or serum sample contains nAbs, cell infection will be inhibited, resulting in a reduced number of viral plaques. As the titer of nAbs in the plasma or serum sample increases, the observed PFU decreases [79].

In clinical trials, the Moderna COVID-19 vaccine mRNA-1273 was evaluated for its neutralizing activity with the PRNT using live SARS-CoV-2 [22]. Owing to the time-intensive nature of PRNT, results of the clinical trial were only reported for the low-dose and high-dose groups on days 1 and 43.

Although PRNT is the standard method for measuring SARS-CoV-2 nAbs, it is time-consuming and unsuitable for the rapid detection of nAbs in clinical trials of emerging, highly pathogenic viral infections. Moreover, plaque identification and manual counting require experienced laboratory personnel who can recognize plaques, and different experimenters may yield variable analysis results. Additionally, it has been found that the SARS-CoV-2 Delta and Mu variants cause cell fusion without plaque formation [80], rendering the PRNT ineffective for their detection.

### 6.3. Recombinant Live Virus

The time-consuming and complex processes of traditional live virus-neutralizing antibody detection methods have several drawbacks, including the need for laboratory personnel with a certain level of experience in virus plaque recognition and manual counting, as well as the use of different operators, which can lead to different analysis results. To overcome these limitations, researchers have generated modified recombinant live viruses by inserting reporter genes, with a good correlation between reporter gene expression and viral replication. Additionally, recombinant viruses exhibit growth kinetics and plaque formation that are comparable to those of wild-type viruses, thereby offering a sensitive and objective alternative to neutralizing antibody assays [81,82,83]. For instance, researchers report a trans-complementation system that produces single-round infectious SARS-CoV-2 that can be safely used at BSL-2 laboratories for high-throughput neutralization and antiviral testing [84]. Following the emergence of SARS-CoV-2, researchers constructed a genetically stable reporter virus (mGFP Δ3678_WA1-spike) by deleting four auxiliary genes of SARS-CoV-2 and combining them with modified green fluorescent protein sequences (mGFP). This highly attenuated SARS-CoV-2 can be safely tested for nAbs in the BSL-2 laboratory [85]. In addition, researchers developed a novel single-round infection fluorescent SARS-CoV-2 virus (SFV) that can be safely used at BSL-2 laboratories for high-throughput neutralization and antiviral testing. The SFV neutralization test (SFVNT) has 100% sensitivity and specificity compared to the PRNT [86]. These recombinant viruses could successfully infect primary airway epithelial cells in culture and generate plaque morphology and growth curves similar to those of wild-type viruses. The neutralizing antibody titers detected in convalescent patient sera using this recombinant live virus system demonstrated comparable results to those of PRNT. This recombinant live virus method was also used for neutralizing antibody detection in phase I/II clinical trials of the BNT162b1 and BNT162b2 vaccines [87,88,89,90]. In this assay, inactivated test sera were incubated with the reporter virus for 1 h at 37 °C, followed by infection of Vero E6 cells pre-seeded in a 96-well plate. The neutralizing antibody titer was determined by detecting the fluorescent foci of virus infection [70].

After cell infection with recombinant live viruses containing reporter genes, fluorescence or luminescence expression can easily be detected. Visualizing the fluorescence intensity using spectrophotometric instruments reduces the time-consuming steps needed with traditional live virus-neutralizing antibody assays. Moreover, this approach offers high sensitivity and automation, making it more suitable for large-scale serological testing [91].

Traditional live virus testing primarily relies on manual interpretation for result determination and estimation, which significantly impacts the objectivity and repeatability of the outcomes. The difficulty in standardizing live virus testing methods is a key factor hindering their widespread adoption. Currently, optimizations in the interpretation of results mainly involve three approaches: virus culture combined with ELISA testing [71], virus culture coupled with quantitative real-time PCR (qRT-PCR) testing [92], and virus culture integrated with automatic cell imaging technology [93,94]. Through the optimization of live virus testing procedures, these modifications have partially enhanced the repeatability and throughput of the testing methods, making preliminary strides towards standardization.

## 7. Pseudovirus Neutralization Test (pVNT)

The biosafety level of SARS-CoV-2 has been classified as level 3 by the WHO. Therefore, research activities involving the live virus require a BSL-3 laboratory. Owing to the high facility and resource requirements of a BSL-3 laboratory and the risk associated with handling live strains, the use of pseudoviruses in BSL-2 laboratories has emerged as an alternative approach.

By anchoring the S protein (the receptor-binding protein of SARS-CoV-2) on the surface of the lentivirus, VSV, or other viruses, a pseudovirus mimicking SARS-CoV-2 can be constructed. This SARS-CoV-2 pseudovirus exhibits a spike (S) protein with a similar structure to that of the live virus but does not require a high-level biosafety laboratory. Additionally, reporter genes, such as GFP, red fluorescent protein, or luciferase, can be inserted in the backbone genome of the pseudovirus. Expression of the reporter gene upon cell infection correlates with the infectiousness of the viral inoculum, enabling convenient and rapid evaluation of viral infection or the neutralizing effect of antibodies. The most commonly used pseudovirus packaging systems include lentivirus (e.g., HIV) vector packaging systems, vesicular stomatitis virus (VSV) packaging systems, murine leukemia virus (MLV) packaging systems, and reverse genetic self-assembly pseudovirus systems [95,96]. Researchers have discussed various methods using SARS-CoV-2 spike-pseudo-typed viruses to assess antibody neutralization. Despite minor differences in how each virus model measures sensitivity, the results consistently correlate well with those from authentic SARS-CoV-2 neutralization assays [97], which are crucial for evaluating the effectiveness of vaccinations and the potency of therapies like convalescent plasma.

### 7.1. HIV Lentivirus System

Among lentivirus vector systems, HIV was the first packaging vector developed and widely used for the preparation of various pseudoviruses [98]. This is also one of the main packaging systems for SARS-CoV-2 pseudoviruses [99], typically achieved by co-transfecting two or three plasmids into cells to produce the pseudovirus. The dual-plasmid system includes one plasmid expressing the SARS-CoV-2 S protein and another plasmid expressing the packaging proteins with the envelope (E) gene deleted from the HIV backbone [100]. The HIV triple-plasmid packaging system typically consists of a packaging plasmid, a transfer plasmid containing the reporter gene, and a plasmid expressing the SARS-CoV-2 S protein [101,102]. Transfection of the plasmids into 293T cells leads to the production of pseudovirus particles with the S protein as the surface protein [103]. Compared with other viral vectors, HIV vectors have a large capacity for accommodating exogenous genes (8–9 kb) [104]. Ren et al. [105] constructed a eukaryotic expression plasmid pcDNA3.1–SARS-CoV-2 S expressing the SARS-CoV-2 S protein and co-transfected it with an HIV-1 pNL4-3.Luc.R-E- packaging plasmid containing the Fluc reporter gene in HEK293T cells, resulting in the production of a SARS-CoV-2 pseudovirus. This pseudovirus can be used for large-scale serological screening in SARS-CoV-2 epidemiological investigations and for evaluating the neutralizing activity of vaccines and therapeutic antibodies (Figure 2a). In clinical applications, the Moderna SARS-CoV-2 mRNA vaccine mRNA-1273 and its bivalent COVID-19 vaccine both used lentivirus-packaged pseudoviruses expressing the full-length S protein of SARS-CoV-2 in neutralization testing [21].

### 7.2. Vesicular Stomatitis Virus (VSV) System

VSV is an enveloped negative-sense RNA virus that can infect various animals and rarely infects humans, causing only mild flu-like symptoms with infection [106]. Previous studies have shown that VSV particles do not exhibit specific selectivity for the type of viral envelope protein incorporated. When cells are co-infected with VSV and other enveloped viruses, VSV can easily become enveloped by the membrane proteins of other viruses [107]. VSV particles can bud without the presence of the G glycoprotein, and the nature of VSV can be altered by the insertion of heterologous glycoproteins. This leads to the development of recombinant VSV (rVSV), in which the gene for the G glycoprotein is replaced with a reporter gene, resulting in rVSV-ΔG, and the envelope protein is derived from the target virus under study [108,109]. Therefore, rVSV-ΔG can be used to produce single-cycle-restricted VSV pseudoviruses with any viral surface glycoprotein in a BSL-3 laboratory. After the outbreak of COVID-19, a research team synthesized the spike (S) gene sequence of SARS-CoV-2, successfully prepared SARS-CoV-2 pseudoviruses based on the VSV packaging system, and established a neutralizing antibody detection method based on SARS-CoV-2 pseudoviruses [66]. Zettl et al. [110] prepared SARS-CoV-2 pseudoviruses using the VSV*ΔG (FLuc) vector, enabling rapid quantification of SARS-CoV-19 nAbs in convalescent COVID-19 patients and vaccinated individuals under BSL-1 conditions (Figure 2b). For the COVID-19 vaccine BNT162b2, derived from German phase I/II trials, the broad-spectrum induction of antibodies by BNT162b2 vaccination was studied using VSV-based SARS-CoV-2 pVNT. The B.1.351 pseudovirus was neutralized at a lower geometric mean titer (GMT) than that of the wild-type strain; all BNT162b2 immune sera showed neutralizing activity against the 22 pseudoviruses tested [20].

This system is efficient and cost-effective, and virus production, titration, and infection assays can be completed within one week, making the system suitable for neutralizing antibody analysis or high-throughput therapeutic screening. However, an important issue with VSV-based pseudoviruses is the presence of residual VSV, which could lead to false-positive results [111].

### 7.3. MLV System

The HIV packaging system and VSV packaging system are the two most commonly used packaging systems for generating SARS-CoV-2 pseudoviruses. In addition to these, the Moloney MLV packaging system has been used in SARS-CoV-2 pseudovirus production. MLV is a typical simple retrovirus that is enveloped and contains positive-sense RNA, encoding three genes for the viral capsid protein, viral enzymes, and envelope protein (Gag, Pol, and Env) [112]. The process of generating pseudoviruses using the MLV system is similar to that of the HIV system, involving transfection of plasmids carrying MLV structural genes (gag and pol) and the gene encoding the heterologous viral envelope protein into cells. Pseudovirus particles carrying the heterologous viral surface protein are secreted in the cell culture medium to obtain a pseudovirus suspension [113]. Zheng et al. [114] prepared pseudovirus particles containing the SARS-CoV-2 S protein using a defective MLV vector plasmid carrying the Fluc reporter gene and a packaging plasmid encoding MLV gag/pol. This pseudovirus can be used under BSL-2 conditions for the specific measurement of neutralizing antibody titers against SARS-CoV-2 in plasma (Figure 2c).

In response to the prevalence and emergence of SARS-CoV-2, many researchers have developed various pseudovirus-based cell culture methods for neutralizing antibody detection. Compared with cVNT, pVNT does not require high-level biosafety laboratories. These methods are safer, simpler, time-saving, and have higher throughput. The inclusion of reporter genes for detection allows for objective result interpretation. Pseudoviruses serve as an alternative and conceptually validated testing approach, with high practical value in rapid clinical vaccine trials during an outbreak. However, this method also requires validation and standardization and should demonstrate comparability with live virus approaches.

## 8. Surrogate Virus Neutralization Test (sVNT)

Whether using a live virus or pseudovirus, there are several common issues in neutralization testing. First, the neutralizing ability of antibodies highly depends on the viral state or titer and the cell type and conditions used in the assay. If the virus and host cells are not in optimal condition for the assay, the reproducibility of the results may be poor [115]. Since the initial outbreak of COVID-19, researchers have developed molecular interaction-based non-virus neutralization antibody detection methods for rapid measurement of SARS-CoV-2 nAbs. Enzyme-linked immunosorbent assay (ELISA), which includes direct, indirect, sandwich, and competitive types, is commonly used for SARS-CoV-2 antibody detection [116]. Studies have shown that different SARS-CoV-2 proteins conjugated with horseradish peroxidase (HRP) can directly bind to the ACE2 receptor, and there is dose-dependent specific binding between ACE2 and the RBD or S1 domain but not with the nucleocapsid (N) protein [67]. Therefore, this method uses the specific binding between the SARS-CoV-2 S protein binding domain (RBD or S1) antigen and the receptor protein ACE2 to mimic virus-host cell interaction. HRP-labeled ACE2 or RBD generates a detection signal, and viral infection can be detected via a colorimetric reaction. When nAbs are present in the test sample, these block the protein interaction, resulting in a weakened colorimetric reaction (Figure 3). The test can typically be completed within a few hours and can be used for qualitative or quantitative measurement [67,117].

Competitive ELISA for SARS-CoV-2 identifies epitopes, including the SARS-CoV-2 S protein, S1 domain, S2 subunit, and RBD domain of the S protein. Competitive ELISA with S1 or RBD-specific binding antibodies is commonly used for neutralizing antibody detection. Because ACE2 has the best binding affinity with RBD, the RBD–ACE2 neutralization activity of serum is nearly equivalent to the virus neutralization activity of its antibodies [67,118]. Therefore, RBD-specific binding antibodies can be used to assess the level of nAbs in individuals immunized with the COVID-19 vaccine. In clinical trials evaluating nAbs after COVID-19 vaccination, ELISA has been used to measure RBD-specific binding antibodies, and the seroconversion rate and GMT of RBD-specific binding antibodies have been reported for vaccines such as CoronaVac, Ad5-nCoV, and ZF2001 [12,30,35].

Competitive ELISA methods can be performed under standard laboratory safety conditions and offer advantages, such as high throughput and short detection time [119]. These methods have been widely used for assessing vaccine efficacy in large-scale clinical trials post-vaccination. However, whether targeting RBD or S1-specific binding antibodies, the currently established ELISA methods are directed against specific regions of the S protein rather than true nAbs. This makes it challenging to predict the neutralizing effects of antibodies out of the RBD or S1 of the virus and their protective efficacy in the host [120].

The Turn-Around Time (TAT) for different neutralizing antibody detection methods varies depending on the type of assay used. Traditional methods, like the cVNT, can take several days due to the need for virus culturing and observation of effects on cell cultures. Notably, the Recombinant live virus assay has seen significant improvement in TAT [121]. In contrast, newer methods, like high-throughput neutralizing antibody assays, which often utilize fluorescent or luminescent reporting, can provide results much more quickly, sometimes within the same day [116]. These faster methods are crucial during pandemics for the timely monitoring of immune responses. Both cVNT and pVNT can provide quantitative results, namely antibody titers, indicating to what extent serum dilution can still effectively neutralize the virus. The sVNT can detect the presence of specific antibodies and provide a relative abundance of antibodies for quantitative application. Different methods have their advantages and disadvantages. When selecting an appropriate detection method, it is necessary to decide which method to use and whether the quantitative analysis is needed, based on the specific needs of the research or diagnosis (Table 2).

## 9. Correlation of Neutralizing Antibodies (nAbs) with a Protective Effect

Both natural infection and vaccination can induce an immune response and the production of nAbs. Finding the correlation between vaccine-induced nAbs and protective efficacy is a crucial and challenging step in vaccine clinical trials [55]. Measurement of virus-neutralizing antibody titers in the blood after COVID-19 vaccination can serve as a surrogate endpoint for evaluating vaccine efficacy [126].

Clinical trial data for different vaccines and real-world data from many countries have shown a positive correlation between neutralizing antibody titers and protective efficacy [62,127,128]. A comparison of serum-neutralizing activity induced by seven major vaccines used in countries worldwide showed that, as long as serum-neutralizing activity reaches 20.2% of the mean convalescent serum-neutralizing activity, more than 50% protection is provided (95% confidence interval (CI) 14.4–28.4%). Furthermore, if the goal is to prevent a 50% chance of severe illness, a neutralizing activity of 3% of the mean convalescent serum neutralizing activity is sufficient (95% CI 0.7–13%, *p* = 0.0004) [62]. In a study conducted in Israel among 1497 healthcare workers who received two doses of vaccination, 39 breakthrough infections (0.34%) occurred. The study found that the GMT of nAbs in breakthrough infections (before infection) was 192.8, compared with 530.4 in non-infected individuals, representing only 36.35% of the latter. Additionally, among breakthrough infections, higher neutralizing antibody levels before infection were associated with lower viral loads. These findings indicate that the level of nAbs induced by COVID-19 vaccines can be used to predict post-vaccination protection [129]. Therefore, neutralizing antibody levels can be used to assess vaccine effectiveness.

Monitoring the decline of COVID-19 vaccine protection over time can be achieved using neutralizing antibody levels. Multiple studies have shown that neutralizing antibody levels increase rapidly within 14–28 days after initial/booster immunization with COVID-19 vaccines but decline within 6–8 months. However, the rate and extent of decline vary significantly among different age groups and vaccine technology platforms. Studies on neutralizing antibody titers among individuals vaccinated with inactivated COVID-19 vaccines have shown that, after receiving a second dose, the positivity rate and titers of nAbs can be maintained at a high level from 11 to 70 days post-vaccination. However, from days 70 to 332, there is a significant decrease in neutralizing antibody titers, with only a 27% positivity rate [130]. A study among 4868 healthcare workers vaccinated with the BNT162b2 vaccine found a rapid decline in neutralizing antibody titers within the first 3 months, with a lower peak level among individuals aged 65 years and above. The decline was more pronounced at 3 months post-vaccination, followed by a slower decline. The study also showed that immunosuppressed individuals had a 70% reduction in neutralizing antibody levels compared with non-immunosuppressed individuals [131,132]. Therefore, in special populations such as older or immunosuppressed individuals, the peak neutralizing antibody levels induced by COVID-19 vaccines are lower than those in younger adults, and the decline in neutralizing antibody levels over time is faster and more significant.

Another important factor contributing to the decrease in vaccine effectiveness is the emergence of SARS-CoV-2 variants. For 3 years since SARS-CoV-2 was first reported, several variants of concern have emerged, many of which are associated with increased transmissibility or some degree of immune escape. The decline in COVID-19 vaccine efficacy is significantly correlated with reduced neutralizing activity against variant strains [133,134,135,136,137,138].

## 10. Correlation Analysis of Testing Methods

In response to the outbreak of SARS-CoV-2, vaccine development institutions and research teams in various countries have used a range of different neutralizing antibody detection methods. However, the differences in assay formats (e.g., live virus, pseudovirus, and ELISA), target antigens (RBD, S1, S, and N), numerical readouts (absorbance optical density/relative light units/plaques/GFP%), and endpoints have led to an inability to directly compare and analyze immunological data. To address this issue, the WHO initiated a collaborative calibration with the first international standard and reference panel for anti-SARS-CoV-2 antibodies [139]. In this collaborative study, 51 laboratories evaluated the suitability of a plasma panel from 11 convalescent SARS-CoV-2 patients as an International Standard for anti-SARS-CoV-2 nAbs using 125 different methods, including ELISA, cVNT/pVNT, flow cytometry-based assays, lateral flow immunoassays, inhibition assays, and dual-antigen binding assays. For live virus assays, except for two low-titer samples and one negative sample for which titers could not be easily calculated, the geometric mean titers (GMTs) of the seven co-calibrated samples were 317.1 (PRNT), 445.3 (focus reduction neutralization test, FRNT), 93.9 (CPE), and 239.6 (microneutralization, MN). For pseudovirus assays, the total GMTs were 371.8 (PV-HIV) and 519.2 (PV-VSV). In a research project to establish the National Standard for anti-SARS-CoV-2 nAbs in China, which can be traced back to the WHO International Standard [140], six laboratories used the live virus PRNT and CPE methods, and four laboratories used the pseudovirus PV-VSV method. The co-calibration results of the two candidate standards showed that using the wild-type strain, the GMTs of samples detected using the live virus method were 133 and 194, and the GMTs of samples detected by the pseudovirus method were 641 and 1512. Using the Delta variant, the GMTs of samples 33 and 66–99 were 62 and 186 with the live virus method, and the GMTs for the pseudovirus method were 289 and 1889, respectively.

Researchers compared and analyzed the serum of hospitalized COVID-19 patients with and without acute respiratory distress syndrome (ARDS) using SARS-CoV-2 PRNT and HIV-based pVNT. The PRNT and pVNT results were comparable. However, in serum testing of patients with ARDS, the titer discrepancies observed in PRNT could not be fully resolved by pVNT, and there was no significant correlation between RBD-binding immunoglobulin G (IgG) measured via ELISA and neutralization titers. In contrast, in the serum of non-ARDS patients, there was a clear correlation between titers of RBD-binding IgG measured with ELISA and the neutralizing activity of these sera [125]. Researchers have found that the correlation between cVTN, sVNT, and pVTN varies among different virus strains, but overall shows a moderate to strong correlation. Specifically, there is a certain degree of correlation between the binding antibody levels detected by sVNT and the neutralizing antibody titers measured by cVNT [141]. Comparisons between cVNT and four alternative neutralization assays based on S-RBD or S1-specific binding (two chemiluminescence assays and two ELISA assays) revealed good agreement between cVNT and sVNT [142]. However, sVNT may only reveal a fraction of nAbs and does not measure total nAbs or neutralizing activity against epitopes outside the RBD, such as the N-terminal domain of the S protein [143]. Therefore, cVNT cannot be replaced in neutralization detection. In the summary data of the phase I/II clinical trial of the inactivated vaccine CoronaVac against SARS-CoV-2 [12], the correlation coefficients between live virus-neutralizing antibody titers and both pseudovirus-neutralizing antibody titers and RBD-specific IgG were greater than 0.8. The correlation coefficient between pseudovirus-neutralizing antibody titers and RBD-specific IgG was 0.73.

These studies indicate that, although different detection methods are based on different technical principles, they can complement each other in the determination of SARS-CoV-2 neutralizing antibodies, providing a certain degree of mutual validation. This is of great significance for evaluating the immune protection or vaccine efficacy after viral infection.

## 11. Standardization of Neutralizing Antibody Detection Methods

The COVID-19 pandemic triggered a global health emergency, and various vaccine development institutions worldwide continue the process of developing candidate vaccines using different platforms targeting different epitopes of SARS-CoV-2. Researchers have evaluated vaccines through in vitro neutralization assays using various infection factors, cell lines, and reporting systems. However, the diversity of testing protocols hinders direct comparisons among the different study results. Neutralizing antibody levels are important indicators of vaccine effectiveness and are critical for therapeutic and sero-epidemiological studies [116,140]. The accuracy, comparability, and reliability of different types of neutralizing antibody assays are of great importance in vaccine development, production, and application. However, owing to the use of different testing methods and laboratories, it is challenging to compare the efficacy of different assay results. Therefore, standardization of neutralizing antibody detection is crucial to ensure the comparability of testing results across different laboratories [144].

Before the availability of internationally standardized substances, qualitative detection of SARS-CoV-2 nAbs was most frequently performed. As research progresses, a clearer understanding of the relationship between levels of SARS-CoV-2 nAbs and vaccine protection is needed. To ensure consistency of SARS-CoV-2 neutralizing antibody results across multiple laboratories and testing methods, the Coalition for Epidemic Preparedness Innovations established a centralized global network in February 2020 for the evaluation and comparison of immune responses induced by candidate vaccines [145]. Using the same assay reagents and standardized protocols to assess vaccine immunogenicity, most inter-laboratory differences were eliminated, enabling head-to-head comparisons of immune responses induced by the various candidate vaccines. In July 2020, the WHO initiated the calibration of international standard substances for SARS-CoV-2. By summarizing the results of 27 neutralization assay methods, the first-generation international standard substance for SARS-CoV-2 antibodies (NIBSC code: 20/136) was released. The specified neutralizing antibody potency of the SARS-CoV-2 international standard substance was determined to be 250 IU per vial. After dissolution in 0.25 mL of distilled water, the final concentration of the preparation is 1000 IU/mL [146]. Subsequently, the SARS-CoV-2 Antibody International Reference Panel (NIBSC code: 20/268) was introduced. Using the concentration unit of 1000 IU/mL assigned in 20/136, the four samples in the international reference panel 20/268 were assigned different antibody concentrations, including neutralizing antibody and IgG concentrations against four different antigens (RBD, S1, N, and S protein) [147]. In December 2020, the WHO announced the International Standard for Anti-SARS-CoV-2 Immunoglobulin (NIBSC code: 21/340). The international standard allows precise calibration of analyses in arbitrary units, reducing inter-laboratory differences and creating a common language for reporting data. The international standard is based on human plasma collected from convalescent patients and freeze-dried in vials, with each vial containing 250 IU as the specified unit of neutralizing activity [148]. All global vaccine developers, national reference laboratories, and academic researchers should use the international standard correctly and report immunogenicity results using international standard units.

However, due to the many complex factors involved, including the characteristics of SARS-CoV-2 that are easy to mutate, the immune system’s response to the virus, and the differences between experimental techniques and populations, the WHO and the international public health authorities did not announce any specific protective level of antibodies for the diseases.

The different types of antibodies detected by different methods, as well as the different sensitivities to the affinity and specificity of antibodies, can affect the accuracy of the detection results. The variation of SARS-CoV-2 may affect the sensitivity and specificity of some detection methods, especially when the antibody target changes. These factors will affect the comparability and interpretation of detection results between different methods.

## 12. Conclusions

In the development and evaluation of vaccines, vaccine efficacy is the most critical assessment parameter. Clinical trial data for different vaccines and real-world data from many countries have consistently shown a positive correlation between neutralizing antibody levels and vaccine protection. Therefore, one of the most important indicators for evaluating vaccine efficacy is the level of nAbs in vaccines. However, the diversity of methods and target antigens used for neutralizing antibody detection in clinical settings creates challenges in comparing results obtained using different methods. The cVNT is considered the gold standard for neutralizing antibody detection and is required in vaccine clinical trials. However, for newly emerging and highly infectious viral diseases, such as SARS-CoV-2, the use of live viruses in neutralization assays is limited by time-consuming procedures and the need for high-level biosafety laboratories. Therefore, pseudovirus-based neutralization assays are commonly used in viral vaccine evaluation. This method offers advantages, such as a high level of safety, simplicity, speed, and high throughput capability. Additionally, specific antigens can be used to detect corresponding nAbs via immune responses, such as with ELISA and other methods.

Multiple studies have shown that higher levels of neutralizing antibodies are often associated with better protection against COVID-19. With the standardization of SARS-CoV-2 neutralizing antibody detection methods and the development of standard substances, the use of nAbs as a vaccine endpoint has been rapidly advanced, promoting virological research and vaccine development for SARS-CoV-2. Researchers have attempted to resolve the differences in the specific antibody levels required for protection by a COVID-19 vaccine through the standardization of antibody titers and the use of more unified analytical methods [149]. However, due to the constantly changing strains of SARS-CoV-2, setting a clear protection threshold is challenging.

## Figures and Tables

**Figure 1 vaccines-12-00554-f001:**
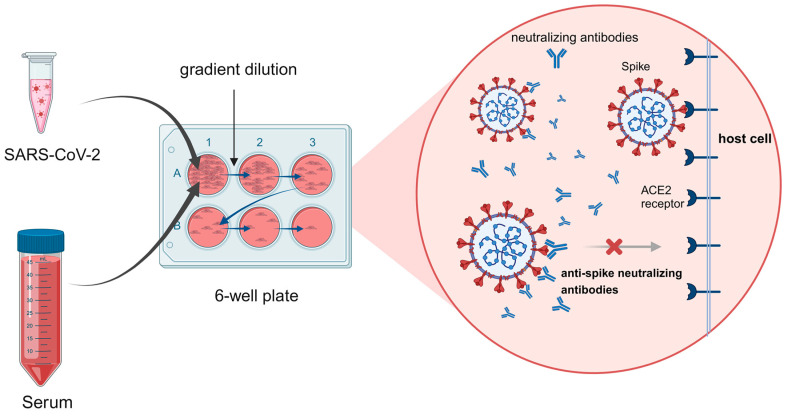
Mechanism of neutralizing antibody detection using live viruses. In the absence of a neutralizing antibody, the SARS-CoV-2 virus binds ACE2, followed by membrane fusion and cell entry; the release of its genetic material leads to virus propagation. The presence of neutralizing antibody blocks ACE2 interaction with the virus. (Figure created with BioRender.com accessed on 27 April 2024).

**Figure 2 vaccines-12-00554-f002:**
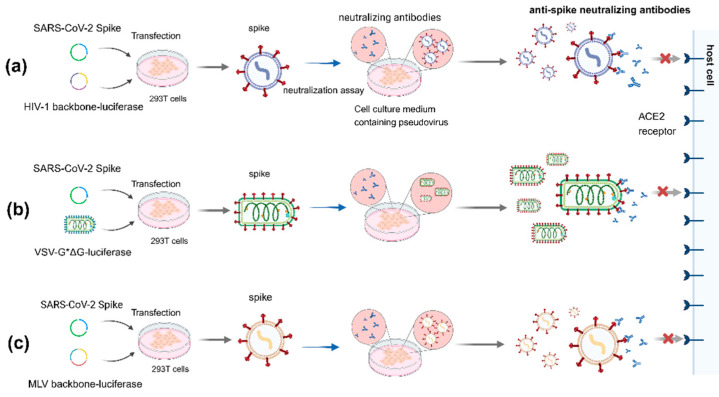
Schematic diagram of neutralizing antibody detection using pseudoviruses. (**a**). The 293T cells are transfected with a plasmid encoding the lentiviral backbone and a plasmid expressing envelope protein. The transfected cells produce recombined pseudoviruses. (**b**). The 293T cells are first transfected with an envelope protein expression plasmid; 24 h post-transfection, the cells are infected with VSV*∆G encoding firefly luciferase. The transfected cells produce recombined pseudoviruses. (**c**). The 293T cells are co-transfected with an envelope protein-encoding plasmid, an MLV packaging transfer vector encoding a luciferase reporter. The transfected cells produce pseudo-typed MLV particles, like the HIV system. SARS-CoV-2 pseudoviruses are capable of binding ACE2 and entering the infected cells; non-replicative pseudovirus binds ACE2 and is internalized. Neutralizing antibody blocks ACE2 interaction with the pseudovirus. (Figure created with BioRender.com accessed on 27 April 2024).

**Figure 3 vaccines-12-00554-f003:**
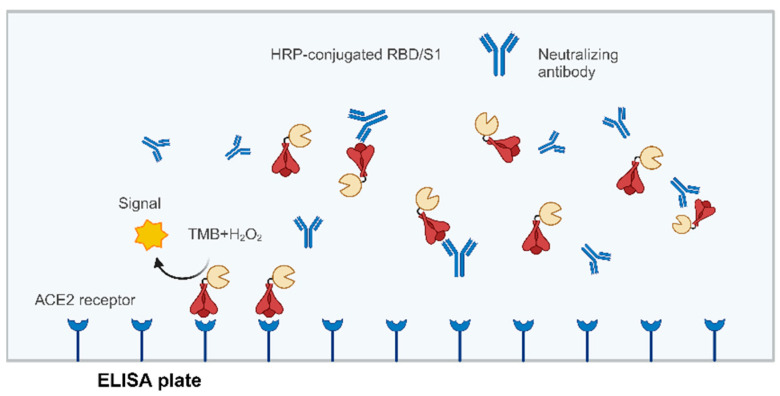
Principle of neutralizing antibody detection using competitive ELISA. The HRP coupling fragment of RBD or S1 binds to samples containing neutralizing antibodies. This mixture is added to an ELISA well plate that has been coated with ACE2 receptors. If the antibodies in the sample have neutralizing antibody activity, the binding between HRP-RBD/S1 and ACE2 will be broken, with cleaning to remove HRP-RBD/S1. The signal generated using a light absorption microplate reader will be lower in the presence of neutralizing antibodies. (Figure created with BioRender.com accessed on 27 March 2024).

**Table 1 vaccines-12-00554-t001:** Main available COVID-19 vaccines.

Vaccine Type	Name	Manufacturer	Immunogenic Detection Method	Immune Indexes	References
Inactivated vaccines	CoronaVac	Sinovac	• Neutralization potency against pseudovirus. • Neutralization potency against live SARS-CoV-2 (CPE). • RBD-specific IgG were detected using the chemiluminescence qualitative kit. • T-cell response was determined with the ELISpot method.	• Seroconversion rates of neutralizing antibodies to live SARS-CoV-2	[12,13,14]
				• Seroconversion rates of binding antibodies to SARS-CoV-2 RBD-specific.	
				• GMT of neutralizing antibodies to live SARS-CoV-2.	
				• GMT of SARS-CoV-2 RBD-specific binding antibodies.	
	BBIBP-CorV	Sinopharm/BIBP	• Neutralization potency against live SARS-CoV-2 (CPE).	• Seroconversion rates of neutralizing antibodies to live SARS-CoV-2	[15,16]
				• GMT of neutralizing antibodies to live SARS-CoV-2.	
mRNA vaccines	BNT162b2	Pfizer-BioNTech	• Neutralization potency against live SARS-CoV-2 (recombinant live virus) • Neutralization potency against pseudovirusVSV-SARS-CoV-2 spike. • SARS-CoV-2-S1–specific IgG/RBD–specific IgG direct Luminex immunoassay	• GMT of SARS-CoV-2 S1-specific binding antibodies.	[17,18,19,20]
				• GMT of neutralizing antibodies to live SARS-CoV-2.	
	mRNA-1273	Moderna	• Neutralization potency against pseudovirus. • Neutralization potency against live SARS-CoV-2 (PRNT). • Binding antibody responses against SARS-CoV-2 S protein were assessed by ELISA. • T-cell responses against the spike protein were assessed by an intracellular cytokine–staining assay.	• GMT of SARS-CoV-2 S-specific binding antibodies.	[21,22]
				• GMT of neutralizing antibodies to pseudovirus.	
				• GMT of neutralizing antibodies to live SARS-CoV-2.	
	mRNA-1273.214	Moderna	• Neutralization potency against SARS-CoV-2 pseudovirus.	• GMT of neutralizing antibodies to the omicron BA.1 variant	[23]
				• GMT of neutralizing antibodies to the omicron BA.4 and BA.5 (BA.4/5)	
Adenovirus vaccines	Ad26.COV2.S	Janssen–Cilag InternationalNV	• SARS-CoV-2 S-specific binding antibodies were measured with ELISA. • SARS-CoV-2 serum neutralizing-antibody titers in a random subgroup of samples by means of a wild-type virus microneutralization assay(wtVMA). • S-specific T-cell responses were assessed by intracellular cytokine staining.	• Seroconversion rates of neutralizing antibodies to live SARS-CoV-2	[15,24]
				• GMC of SARS-CoV-2 S-specific binding antibodies.	
				• GMT of neutralizing antibodies to live SARS-CoV-2.	
	AZD1222/Vaxzevria	AstraZeneca	• SARS-CoV-2 S-specific binding antibodies were measured with ELISA. • Neutralization potency against live SARS-CoV-2 by microneutralization assay. • Neutralization potency against live SARS-CoV-2 • T-cell response was determined with the ELISpot method.	• SARS-CoV-2 S-specific binding antibody titers	[25,26,27]
				• GMT of neutralizing antibodies to live SARS-CoV-2.	
	Sputnik V	Gamaleya Nat. Center of Epidem. and Microbiol.	• SARS-CoV-2 RBD-specific IgGs were measured with ELISA. • The titer of nAbs was measured by microneutralization assay. • Antigen-specific proliferating CD4 and CD8 cells by flow cytometry	• The change from baseline in antigen-specific antibody levels.	[28,29]
				• SARS-CoV-2 S-specific binding antibody titers	
				• Specific T-cell immunity and interferon-γ production or lymphoproliferation	
	Ad5-nCoV/Convidecia	CanSinoBIO	• SARS-CoV-2 RBD-specific binding antibodies were measured with ELISA. • Neutralization potency against live SARS-CoV-2 (recombinant live virus) • Neutralization potency against pseudovirusVSV-SARS-CoV-2 spike. • T-cell response was determined with the ELISpot method.	• GMT of neutralizing antibodies to live SARS-CoV-2.	[30,31]
				• GMT of neutralizing antibodies to pseudovirus.	
				• The RBD-specific ELISA antibodies peaked.	
				• Seroconversion rates of neutralizing antibodies to live SARS-CoV-2	
				• Seroconversion rates of neutralizing antibodies to pseudovirus.	
				• Seroconversion rates of RBD-specific binding antibodies.	
Subunit vaccines	NVX-CoV2373/Nuvaxovid	Novavax	• IgG anti–spike protein response (ELISA) • Wild-type virus neutralization assays(MN) • Antigen-specific CD4+ T cells was determined with the intracellular cytokine staining	• ELISA anti-spike IgG geometric mean ELISA units (GMEUs)	[32,33]
				• Wild-type SARS-CoV-2 microneutralization assay at an inhibitory concentration greater than 99% (MN IC > 99%) titer responses	
	ZF2001	Zhifei Longcom	• SARS-CoV-2 RBD-specific binding antibodies were measured with ELISA. • Neutralization potency against live SARS-CoV-2 (CPE) • T-cell responses against the spike protein were assessed by an enzyme-linked immuno-spot (ELISpot)	• Seroconversion rates of RBD-binding IgG	[34,35]
				• GMTs of RBD-binding antibodies	
				• Seroconversion rates of neutralizing antibodies to live SARS-CoV-2.	
				• GMTs of neutralizing antibodies to live SARS-CoV-2.	

**Table 2 vaccines-12-00554-t002:** Advantages and disadvantages of different neutralizing antibody detection methods.

Method Classification	Method Name	Advantages	Disadvantages
cVNT	CPE	• The “gold standard” for neutralizing antibody detection has specificity, sensitivity, and can accurately measure the neutralizing ability of antibodies [76].	• BSL-3, low throughput, time-consuming, high condition requirements, and is influenced by subjective explanations from researchers, resulting in cumbersome steps [76,122].
	PRNT		
	Recombinant live virus	• Reduced time consumption, dynamic monitoring, and objective results [70]. • Can be safely used at BSL-2 laboratories [84,85].	• Cumbersome and time-consuming steps [84,85].
pVNT	HIV system	• BSL-2, high throughput, less time-consuming, specific, sensitive, consistent with live virus results, accurately measure the neutralizing ability of antibodies [66]; • Lentiviral vector-based pseudoviruses differ from conventional retroviral vectors in that they have the ability to infect both dividing and non-dividing cells; • VSV systems are suitable for neutralizing antibody analysis or screening of high-content therapies [123].	• The number of envelope proteins in pseudoviruses is not directly proportional to the copy number of the core genome. The titer needs to be measured through qPCR and other methods, which loses its authenticity, especially when comparing the impact of different mutant S proteins on viral infectivity [105]; • VSV residue in the VSV system may lead to false positive results [111]; • Comparison and validation with live viruses are required.
	VSV system		
	MLV system		
sVNT	ELISA	• Cell-independent, safe; high throughput, low cost, less time-consuming [119]; can be used for screening for neutralizing antibodies in the population to investigate the efficacy of protective immunity and vaccination [124].	• Does not represent actual neutralizing antibody titers, does not provide information on the neutralizing ability of antibodies and has low sensitivity for early infection [67,125].

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
