# Peer review of "SARS-CoV-2 Neutralization Assays Used in Clinical Trials: A Narrative Review"

_vaccines, 2024, doi:10.3390/vaccines12050554_

Round 1
Reviewer 1 Report
Comments and Suggestions for Authors
This is a valuable and comprehensive manuscript, which aims at reviewing and discussing the neutralizing antibodies (nAbs) against SARS-CoV-2 and the related detection methods. As nAbs are also important indicators for the effectiveness of SARS-CoV-2 vaccines, the authors also emphasize that standardization of nAb detection methods are needed to evaluate antiviral products. The current stage of the method standardization is further discussed at the end. Thus, the manuscript is of interest to the readers in this field, and I only have a few suggestions:
Major points:
1) This manuscript lacks a brief review of both humoral and cellular immune responses against SARS-CoV-2 infection, otherwise it will mislead the readers that B cells mediated humoral immunity is the only antiviral immune response.
2) Since the different study results of Live virus neutralization test were generated from different laboratories, could the authors please comment on the standardization of this detection method.
3) A table is needed to compare the advantages and disadvantages of different detection methods.
4) Could the authors please specify the effectiveness of each SARS-CoV-2 vaccine in Table 1?
Minor points:
Line 31, “leading to” should be replaced by “followed by”.
Line 32, the human body generates antibodies immediately after infection, and the antibody levels may last for up to several months. Thus, the description here is not accurate.
Line 35, The sentence of “Individuals infected with SARS-CoV-2 as well as those who have received vaccination against the virus can produce nAbs” should be deleted. It has the same meaning with the sentence in line 32 and is not necessary to repeated here.
Line 37, “monitoring of viral infections” should be “monitoring the viral infections”.
Line 42-44, I’m able to understand what the authors’ saying, but the readability of the text is not good.
Line 49, “product introduction” could be deleted.
Lihe 51, “existing” could be deleted.
Table 1, the abbreviations are defined.
Line 54, references supporting that the safety and efficacy of the vaccines have been validated are missing.
Line 61, “kill” should be replaced by “inactivate”.
Line 81, the revised sentence is “inserting specific antigenic nucleotide sequence into viral vectors to express the antigens of interest in host cells”.
Line 90, refer to “host translation machinery to produce the antigen protein”.
Line 123, refer to “all the approved vaccines on the WHO emergency use list”
Comments on the Quality of English Language
Mionr editing is necessary.
Author Response
1) This manuscript lacks a brief review of both humoral and cellular immune responses against SARS-CoV-2 infection, otherwise it will mislead the readers that B cells mediated humoral immunity is the only antiviral immune response.
Response: We would like to start by thanking you for the focused guidance about how we can improve our manuscript. We have followed suggestions in this comment and have now comprehensively reworked the manuscript. We have added a brief review of both humoral and cellular immune responses against SARS-CoV-2 infection in line 34-42. It reads now “After vaccination or viral infection, the innate/non-specific immune system is initially activated. Antigen-presenting cells, including macrophages and dendritic cells, engulf the exogenous antigens. Subsequently, antigen-presenting cells transmit antigen in-formation to T cells and B cells, activating the adaptive/specific immune system. Acti-vated CD4+ T cells secrete various cytokines to further stimulate immune responses, and CD8+ T cells eliminate virus-infected cells [5]. B cells are also activated to produce specific antibodies, including neutralizing antibodies (nAbs) and binding antibodies [6]. Therefore, nAbs, cellular immunity, innate immunity, and other immunological indicators are vital in assessing the effectiveness of viral vaccines [7,8]. The importance of measuring nAbs in COVID-19 relies on the fact that they can be used as correlate of protection.”
2) Since the different study results of Live virus neutralization test were generated from different laboratories, could the authors please comment on the standardization of this detection method.
Response: Thanks for your valuable comments. Traditional live virus testing primarily relies on manual interpretation for result determination and estimation, which significantly impacts the objectivity and repeat-ability of the outcomes. The difficulty in standardizing live virus testing methods is a key factor hindering their widespread adoption. We have added this in the revised manuscript in line 314-321. It reads now “ The difficulty in standardizing live virus testing methods is a key factor hindering their widespread adoption. Currently, optimizations in the interpretation of results mainly in-volve three approaches: virus culture combined with ELISA testing[72], virus culture cou-pled with quantitative real-time PCR (qRT-PCR) testing[93], and virus culture integrated with automatic cell imaging technology[94,95]. Through the optimization of the live virus testing procedures, these modifications have partially enhanced the repeatability and throughput of the testing methods, making preliminary strides towards standardization. ”
3) A table is needed to compare the advantages and disadvantages of different detection methods.
Response: Many thanks for your suggestions. Table 2 lists the advantages and disadvantages of different detection methods.
4) Could the authors please specify the effectiveness of each SARS-CoV-2 vaccine in Table 1?
Response: Appreciate your suggestions. The safety and efficacy of the SARS-CoV-2 vaccines have been extensively evaluated globally. However, the effectiveness of these vaccines may vary among different populations and in the presence of variant viruses. Because different types of vaccines have been evaluated in different population confronting different variants in their clinical trials, it is impossible to compare them directly and might implicate misleading conclusions. We have explained this point after Table 1.
Minor points:
Line 31, “leading to” should be replaced by “followed by”.
Response: Appreciate your valuable comments. We have changed it to "followed by " . in line 30.
Line 32, the human body generates antibodies immediately after infection, and the antibody levels may last for up to several months. Thus, the description here is not accurate.
Response: Thanks for your comments. We have changed it in line 31 of the revised manuscript.
Line 35, The sentence of “Individuals infected with SARS-CoV-2 as well as those who have received vaccination against the virus can produce nAbs” should be deleted. It has the same meaning with the sentence in line 32 and is not necessary to repeated here.
Response: Following your suggestions, we have deleted the description in the revised manuscript.
Line 37, “monitoring of viral infections” should be “monitoring the viral infections”.
Response: Following your suggestions, we have corrected it in the revised manuscript.
Line 42-44, I’m able to understand what the authors’ saying, but the readability of the text is not good.
Response: Apologize for the confusing description. We have rephrased it. It now reads “Research and development efforts regarding vaccines against SARS-CoV-2 are ongoing, especially for broadly protective candidate vaccines. Therefore, continuous monitoring is necessary to track emerging variants of SARS-CoV-2 to facilitate the development of the next-generation vaccines.” in line 47-50 of the revised manuscript.
Line 49, “product introduction” could be deleted.
Response: Appreciate your comments. We have deleted in line 55 of the revised manuscript.
Line 51, “existing” could be deleted.
Response: Thanks for your comments. We have deleted in line 57 of the revised manuscript.
Table 1, the abbreviations are defined.
Response: Many thanks for your suggestions. We have added the defines after Table 1 of the abbreviations.
Line 54, references supporting that the safety and efficacy of the vaccines have been validated are missing.
Response: Appreciate your comments. We have added the references in line 63 of the revised manuscript.
Line 61, “kill” should be replaced by “inactivate”.
Response: Thanks for your comments. We have replaced by “inactivate” in line 70 of the revised manuscript.
Line 81, the revised sentence is “inserting specific antigenic nucleotide sequence into viral vectors to express the antigens of interest in host cells”.
Response: Many thanks for your suggestion. We have changed it in line89-90 of the revised manuscript.
Line 90, refer to “host translation machinery to produce the antigen protein” .
Response: Appreciate your suggestion. We have changed it in line106-107 of the revised manuscript.
Line 123, refer to “all the approved vaccines on the WHO emergency use list”
Response: Many thanks for your suggestions. We have changed it in line139 of the revised manuscript.
Reviewer 2 Report
Comments and Suggestions for Authors
Nie et al. conducted a thorough review of three major SARS-CoV-2 neutralizing antibody detection methods. Overall, the manuscript is well-written and demonstrates strong clarity and coherence. However, there are notable areas for enhancement, particularly regarding the section discussing cVNT utilizing recombinant live virus. Updated information, such as the use of attenuated virus in cVTN, should be included. It's important to note that it has been authorized for BSL-2 laboratory settings and successfully employed in neutralization tests. The inclusion significantly advances throughput/turnaround time (TNT), a point warranting emphasis.
Please cite doi.org/10.1038/s41467-020-19055-7 along with ref. #64
Line 292: Please cite DOI: 10.1056/NEJMoa2211031, DOI: 10.1056/NEJMc2113468, doi.org/10.1038/s41586-021-03693-y, DOI: 10.1056/NEJMc2106083, doi.org/10.1038/s41586-020-2814-7, doi.org/10.1038/s41586-020-2639-4
It’s crucial to update the discussion on the BSL-2 level recombinant live virus assay, a big advancement from the BSL-3 essential requirement. Please cite doi.org/10.3390/v15091855, doi.org/10.1016/j.cell.2021.02.044, doi.org/10.3390/v14061211, doi.org/10.1038/s41467-022-31930-z
Additionally, the manuscript should address the TAT for each method discussed. Notably, the Recombinant live virus assay has seen significant improvement in TAT, a point that merits detailed elaboration. Please cite doi.org/10.1016/j.jim.2021.113060
Furthermore, please discuss which method(s) can provide the titers indicating whether the assay is quantitative or qualitative, along with the advantages/benefits of the quantitative assay.
Author Response
Response: We would like to start by thanking you for the focused guidance about how we can improve our manuscript. We have followed the suggestions in this comment and have now comprehensively reworked the manuscript. in line 285-294 of the revised manuscript.
Please cite doi.org/10.1038/s41467-020-19055-7 along with ref. #64
Response: Following your suggestions, we have cited doi.org/10.1038/s41467-020-19055-7 along with ref. #64 ref of the revised manuscript.
Line 292: Please cite DOI: 10.1056/NEJMoa2211031, DOI: 10.1056/NEJMc2113468, doi.org/10.1038/s41586-021-03693-y, DOI: 10.1056/NEJMc2106083, doi.org/10.1038/s41586-020-2814-7, doi.org/10.1038/s41586-020-2639-4
Response: Following your suggestions, we have cited in Line 303 of the revised manuscript.
It’s crucial to update the discussion on the BSL-2 level recombinant live virus assay, a big advancement from the BSL-3 essential requirement. Please cite doi.org/10.3390/v15091855, doi.org/10.1016/j.cell.2021.02.044, doi.org/10.3390/v14061211, doi.org/10.1038/s41467-022-31930-z
Response: Appreciate your suggestions. We have updated the discussion on the BSL-2 level recombinant live virus assay in line 287-297. It reads now “ For instance, researchers report a trans-complementation system that produces single-round infectious SARS-CoV-2 that can be safely used at BSL-2 laboratories for high-throughput neutralization and antiviral testing[85]. Following the emergence of SARS-CoV-2, researchers engineered constructed a genetically stable reporter virus (mGFP Δ3678_WA1-spike) by deleting four auxiliary genes of SARS-CoV-2 and combining them with modified green fluorescent protein sequences (mGFP), this highly attenuated SARS-CoV-2 can be safely tested for nAbs in the BSL-2 laboratory[86]. researchers developed a novel single-round infection fluorescent SARS-CoV-2 virus (SFV) that can be safely used at BSL-2 laboratories for high-throughput neutralization and antiviral testing. The SFV neu-tralization test (SFVNT) has 100% sensitivity and specificity compared to the PRNT.”
Additionally, the manuscript should address the TAT for each method discussed. Notably, the Recombinant live virus assay has seen significant improvement in TAT, a point that merits detailed elaboration. Please cite doi.org/10.1016/j.jim.2021.113060
Response: Appreciate your valuable comments. We have added a discussion about different methods of TAT in line 469-472. It reads now “The Turn-Around Time (TAT) for different neutralizing antibody detection methods varies depending on the type of assay used. Traditional methods like the cVNT can take several days due to the need for virus culturing and observation of effects on cell cultures. Notably, the Recombinant live virus assay has seen significant improvement in TAT.”
Furthermore, please discuss which method(s) whether the assay is quantitative or qualitative, along with the advantages/benefits of the quantitative assay.
Response: Appreciate your valuable comments. We have added different methods to determine whether they are quantitative or qualitative, and the advantages/benefits of the quantitative assay in line476-479 and Table2. It reads now “ Both cVNT and pVNT can provide quantitative results, namely antibody titers, indicating to what extent serum dilution can still effectively neutralize the virus. The sVNT can detect the presence of specific antibodies and provide relative abundance of antibodies for quan-titative application. Different methods have their advantages and disadvantages.
Reviewer 3 Report
Comments and Suggestions for Authors
Dear authors please find my commends below.
I would like to thank you for the opportunity to review the manuscript entitled “SARS-CoV-2 neutralization assays used in clinical trials: a narrative review” For the Vaccines Journal.
The paper is interesting and deal with the recent topic. The introduction is clear and well arranged. The methodology sounds good. The discussion is good even could be improved. The study design is proper and the finding is important for the field. I have following suggestions which should be addressed first.
My decision is Accept after major revisions.
Abstract lines 12-26: add the aim of the review.
Introduction lines 29-47.
Please insert if it is possible a method of sources was used by the authors for the present review. Regardless of the type of narrative review, authors should clearly describe how analyses were conducted and provide justification for their approach.
Lines 52-on the table 1:
Suggested to the authors for better representative and globally scientific approach to add the Sputnik Vaccine by Gamaleya Center.
For the assistance of the authors, I suggest the follow references:
· Longitudinal Study after Sputnik V Vaccination Shows Durable SARS-CoV-2 Neutralizing Antibodies and Reduced Viral Variant Escape to Neutralization over Time. mBio. 2022 Feb 22;13(1): e0344221. doi: 10.1128/mbio.03442-21. Epub 2022 Jan 25.
· Neutralizing activity of Sputnik V vaccine sera against SARS-CoV-2 variants. https://www.nature.com/articles/s41467-021-24909-9 .
· Olga Matveeva, and Alexander Ershov. Retrospective Cohort Study of the Effectiveness of the Sputnik V and EpiVac Corona Vaccines against the SARS-CoV-2 Delta Variant in Moscow (June–July 2021). Vaccines (Basel). 2022 Jul; 10(7): 984.Published online 2022 Jun 21. doi: 10.3390/vaccines10070984.
Lines 550-580: WHO and the international public health authorities didn’t announce any protective level of antibodies of the diseases: I think if the authors could try to explain in the last paragraph why this happen, could be enhanced the total quality of the manuscript. Suggested to use the paradigm of hepatitis b where the 10 IU/ML considered such a protective for the persons where completed the series of vaccination.
Lines 582:
Please insert the main limitations of the study
Conclusions lines 583-601.
Please insert the offer of the present narrative review to international literature (in one paragraph) what new bring?
Please see the follow References and how they could be help you.
· Khoury DS, Schlub TE, Cromer D, Steain M, Fong Y, Gilbert PB, Subbarao K, Triccas JA, Kent SJ, Davenport MP. Correlates of Protection, Thresholds of Protection, and Immunobridging among Persons with SARS-CoV-2 Infection. Emerg Infect Dis. 2023 Feb;29(2):381-388. doi: 10.3201/eid2902.221422.
Author Response
Abstract lines 12-26: add the aim of the review.
Response: Many thanks for your suggestions. We have added the aim of the review: “In this comprehensive review, we discuss the principles, advantages, limitations and application of different SARS-CoV-2 neutralization assays in vaccine clinical trials. Provide guidance for the development and detection of COVID-19 vaccines.” in line 23-25 of the revised manuscript.
Introduction lines 29-47.
Please insert if it is possible a method of sources was used by the authors for the present review. Regardless of the type of narrative review, authors should clearly describe how analyses were conducted and provide justification for their approach.
Response: Follow your suggestion, we have added the method sources in line 50-53. It reads now “In this review, we searched PubMed from January, 2020 to April 29, 2024 to identify all the eligible studies to comprehensively describe the progress in SARS-CoV-2 neutralization assays used in clinical trials.”
Lines 52-on the table 1:
Suggested to the authors for better representative and globally scientific approach to add the Sputnik Vaccine by Gamaleya Center.
For the assistance of the authors, I suggest the follow references:
Longitudinal Study after Sputnik V Vaccination Shows Durable SARS-CoV-2 Neutralizing Antibodies and Reduced Viral Variant Escape to Neutralization over Time. mBio. 2022 Feb 22;13(1): e0344221. doi: 10.1128/mbio.03442-21. Epub 2022 Jan 25.
Neutralizing activity of Sputnik V vaccine sera against SARS-CoV-2 variants. https://www.nature.com/articles/s41467-021-24909-9 .
Olga Matveeva, and Alexander Ershov. Retrospective Cohort Study of the Effectiveness of the Sputnik V and EpiVac Corona Vaccines against the SARS-CoV-2 Delta Variant in Moscow (June–July 2021). Vaccines (Basel). 2022 Jul; 10(7): 984.Published online 2022 Jun 21. doi: 10.3390/vaccines10070984.
Response: We would like to start by thanking you for the focused guidance about how we can improve our manuscript. We have followed suggestions in this comment and have now comprehensively reworked the manuscript. “Sputnik V is one of the first COVID-19 vaccines approved for emergency use, developed by the Gamalea Research Center in Russia, and the first non-Western vaccine to complete all phase III clinical trials [45]. Sputnik V uses two different adenoviral vectors (rAd26 and rAd5) to transmit the S protein gene of the virus, which can enhance the immune system's response and improve the effectiveness and duration of the vaccine[46]. Multiple studies have shown that Sputnik V has shown high efficacy in preventing symptomatic and se-vere infections of COVID-19[10,47].” in line 97-104 of the revised manuscript.
Lines 550-580: WHO and the international public health authorities didn’t announce any protective level of antibodies of the diseases: I think if the authors could try to explain in the last paragraph why this happen, could be enhanced the total quality of the manuscript. Suggested to use the paradigm of hepatitis b where the 10 IU/ML considered such a protective for the persons where completed the series of vaccination.
Response: Appreciate your valuable comments. We have added the information about why WHO and the international public health authorities didn’t announce any protective level of antibodies of the diseases in 629-632. It reads now “But due to many complex factors are involved, including the characteristics of COVID-19 that is easy to mutate, the immune system's response to the virus, and the differences between experimental techniques and populations.”
Lines 582: Please insert the main limitations of the study
Response: Thanks for your comments.. We have added the main limitations of the study in 633-638. It reads now “The different types of antibodies detected by different methods, as well as the different sensitivities to the affinity and specificity of antibodies, can affect the accuracy of the detec-tion results. The variation of SARS-CoV-2 may affect the sensitivity and specificity of some detection methods, especially when the antibody target changes. These factors will affect the comparability and interpretation of detection results between different methods.”
Conclusions lines 583-601.
Please insert the offer of the present narrative review to international literature (in one paragraph) what new bring?
Please see the follow References and how they could be help you.
- Khoury DS, Schlub TE, Cromer D, Steain M, Fong Y, Gilbert PB, Subbarao K, Triccas JA, Kent SJ, Davenport MP. Correlates of Protection, Thresholds of Protection, and Immunobridging among Persons with SARS-CoV-2 Infection. Emerg Infect Dis. 2023 Feb;29(2):381-388. doi: 10.3201/eid2902.221422.
Response: We would like to start by thanking you for the focused guidance about how we can improve our manuscript. We have followed suggestions in this comment and have now comprehensively reworked the manuscript. We added the sentence at the ending as “ The researchers tried to solve the difference between the specific values of antibody levels required for the protection of COVID-19 vaccine by standardizing antibody titers and using more unified analytical methods[147]. However, due to the constantly changing SARS-CoV-2 strains, setting a clear protection threshold is challenging.”
Reviewer 4 Report
Comments and Suggestions for Authors
The review article written by Sun et al. focusses on SARS-CoV-2 neutralization assays performed to evaluate the efficacy of vaccines in clinical trials. To combat the SARS-CoV-2 pandemic, several vaccines including mRNA, subunit, viral vector and inactivated vaccines have been developed. Neutralizing antibody titers have been determined to assess vaccine efficacy. However, neutralizing antibodies can be measured by various methods using live virus, pseudoviruses or surrogate virus. Due to the variations in methodology, it is not possible to compare results obtained from different studies. Therefore, standardized protocols are required to enable comparison of vaccine efficacy among individual studies.
The article is well written und contains all important information and references. The following points have to be addressed prior acceptance for publication:
1. Table 1: Please add a legend to explain the abbreviations GMT and GMC. Why is Sinovac written in bold letters?
2. Figure 1: There is one typing error (neeutralization)
3. Figure legend 2, line 393. (1) should be read (a)
Author Response
- Table 1: Please add a legend to explain the abbreviations GMT and GMC. Why is Sinovac written in bold letters?
Response: Many thanks for your suggestions. We have given the full description of GMT (Geometric mean antibody titer) and GMC(Geometric mean concentration) in line 60 when it appears for the first time in the revised manuscript." Apologize for the Sinovac written in bold letters, and we have already modified this point.
- Figure 1: There is one typing error (neeutralization)
Response: Thanks for finding the typing error in Figure 1, We have reworked this error.
- Figure legend 2, line 393. (1) should be read (a)
Response: Thanks for finding this error in Figure 2. We have now carefully reworked this and now indicate the corresponding figures.
Reviewer 5 Report
Comments and Suggestions for Authors
The authors performed a narrative review on the three different method that has been develop to assess the levels of neutralizing antibodies (nAbs) against SARS-CoV-2, induced by the vaccines against this virus. The information reported is important. Some aspects should be addressed before considering publishing this manuscript.
11. The manuscript should benefit for an edition for English accuracy, at least to avoid redundancy. For example, in Abstract, lines 13-14: the sentence use 4 times the term vaccines.
22. The importance of measuring nAbs in COVID-19 relies on the fact that they can be used as correlate of protection. This should be described in the Introduction. See as an example: Regev-Yochay G, et al. Lancet Microbe. 2023 May;4(5):e309-e318. doi: 10.1016/S2666-5247(23)00012-5.
33. The aim of the study should be included in Introduction.
44. Table 1: What was the criteria for including COVID-19 in the table? OMS Approval? Some of the vaccines that were used (many doses around the world) are not listed, as for example Sputnik V one. The title should be changed or, more advisably, the vaccines most used during the pandemic should all be included.
55. Figure 1 is too small, and Figure 2 could beneficiate of a larger letter size.
66. Table 2. Not detecting nAbs against non-structural proteins (NSPs) is presented as a limitation of pVNT, while eventual nAbs against NSPs is not induced by none of the currently available vaccines, and the focus of this review is based on nAbs induced by vaccines.
77. Again, the focus of this narrative review is on methods for detecting nAbs. The description of the vaccines might then be reduced, and, on the contrary, the authors could include instead, in the section of correlations of neutralization methods, more examples of comparison of the different methodologies.
Comments on the Quality of English LanguageSome minor editing required.
Author Response
- The manuscript should benefit for an edition for English accuracy, at least to avoid redundancy. For example, in Abstract, lines 13-14: the sentence use 4 times the term vaccines.
Response: Thanks for your suggestions. We have revised the manuscript with the help of a native English expert. And we have changed line13-14 to a concise expression specifically.
- The importance of measuring nAbs in COVID-19 relies on the fact that they can be used as correlate of protection. This should be described in the Introduction. See as an example: Regev-Yochay G, et al. Lancet Microbe. 2023 May;4(5):e309-e318. doi: 10.1016/S2666-5247(23)00012-5.
Response: Thanks for your valuable comments. We have added this section describes in line 39-41 .It reads now “The importance of measuring nAbs in COVID-19 relies on the fact that they can be used as correlate of protection”.
- The aim of the study should be included in Introduction.
Response: Many thanks for your suggestions. We have added the aim of the review: “In this comprehensive review, we discuss the principles, advantages, limitations and application of different SARS-CoV-2 neutralization assays in vaccine clinical trials. Provide guidance for the development and detection of COVID-19 vaccines.” in line 23-25 of the revised manuscript.
- Table 1: What was the criteria for including COVID-19 in the table? OMS Approval? Some of the vaccines that were used (many doses around the world) are not listed, as for example Sputnik V one. The title should be changed or, more advisably, the vaccines most used during the pandemic should all be included.
Response:Thanks for the positive appraisal of our study and for the valuable suggestion about how to improve our manuscript. The criteria for inclusion in the table of vaccines are based on the vaccines that were initially widely launched for use. We appreciate this suggestion and have now added Sputnik V on the original basis in table 1 and line 97-104 . It reads now “Sputnik V is one of the first COVID-19 vaccines approved for emergency use, developed by the Gamalea Research Center in Russia, and the first non Western vaccine to complete all phase III clinical trials. Sputnik V uses two different adenoviral vectors (rAd26 and rAd5) to transmit the S protein gene of the virus, which can enhance the immune system's response and improve the effectiveness and duration of the vaccine. Multiple studies have shown that Sputnik V has shown high efficacy in preventing symptomatic and severe infections of COVID-19.”
- Figure 1 is too small, and Figure 2 could beneficiate of a larger letter size.
Response: Following your valuable suggestions, we have redesigned the figure 1 and 2.
- Table 2. Not detecting nAbs against non-structural proteins (NSPs) is presented as a limitation of pVNT, while eventual nAbs against NSPs is not induced by none of the currently available vaccines, and the focus of this review is based on nAbs induced by vaccines.
Response: Thanks for your valuable comments. We have delete this view in Table 2 and changed the disadvantages of pVNT: “The number of envelope proteins in pseudoviruses is not directly proportional to the copy number of the core genome. The titer needs to be measured through qPCR and other methods, which loses its authenticity, especially when comparing the impact of different mutant S proteins on viral infectivity.
- Again, the focus of this narrative review is on methods for detecting nAbs. The description of the vaccines might then be reduced, and, on the contrary, the authors could include instead, in the section of correlations of neutralization methods, more examples of comparison of the different methodologies.
Response: Thanks for prompting us to consider these ideas in detail. We have added more examples of comparison of the different methodologies in line: “566-570 Researchers have fond that th correlation between cVTN, sVNT and pVTN varies among different virus strains, but overall shows a moderate to strong correlation. Specifically, there is a certain degree of correlation between the binding antibody levels detected by sVNT and the neutralizing antibody titers measured by cVNT.”
Round 2
Reviewer 2 Report
Comments and Suggestions for Authors
Line 290: Delete either "engineered" or "constructed". Cannot use both.
Line 293: Start with a new sentence "This highly attenuated..."
Line 294 - 296: Please rewrite this sentence and make it more fluent with the previous sentence.
Line 480: "." should be after the (Table 2).
Table 2: The statement "Less sensitive than PRNT" is not true. Actually, it is more sensitive than PRNT.
Comments on the Quality of English LanguageAuthors should read the manuscript a couple more times to edit certain sentences and punctuations to make them clearer. Only a few are mentioned above.
Author Response
Line 290: Delete either "engineered" or "constructed". Cannot use both.
Response: Many thanks for your suggestions. We have already deleted the "engineered" in line 290 of the revised manuscript.
Line 293: Start with a new sentence "This highly attenuated..."
Response: Following your suggestions, we have changed it to “ This highly attenuated...” in line 293 of the revised manuscript.
Line 294 - 296: Please rewrite this sentence and make it more fluent with the previous sentence.
Response: Follow your suggestions. We have changed it to "In addition, researchers developed a novel single-round infection fluorescent SARS-CoV-2 virus (SFV) that can be safely used at BSL-2 laboratories for high-throughput neutralization and antiviral testing. " in line 294-296 of the revised manuscript.
Line 480: "." should be after the (Table 2).
Response: Many thanks for your suggestions. We have refined it.
Table 2: The statement "Less sensitive than PRNT" is not true. Actually, it is more sensitive than PRNT.
Response: Following your suggestions, we have deleted the description in Table 2 of the revised manuscript.
Comments on the Quality of English Language
Authors should read the manuscript a couple more times to edit certain sentences and punctuations to make them clearer. Only a few are mentioned above.
Response: We would like to start by thanking you for the focused guidance about how we can improve our manuscript. We have followed suggestions in this comment and have now comprehensively reworked the manuscript.
Reviewer 3 Report
Comments and Suggestions for Authors
Dear authors thank you for the revisions.
All my concerns answered .
Author Response
Thank you for your positive feedback on the revised manuscript.
Reviewer 5 Report
Comments and Suggestions for Authors
The authors addressed satisfactorely the concerns.
Author Response

(The authors gave the same response as above.)
